# Reasoning about Uncertainties in Discrete-Time Dynamical Systems using Polynomial Forms.

**Sriram Sankaranarayanan**
University of Colorado Boulder, USA
srirams@colorado.edu

**Yi Chou**
University of Colorado Boulder, USA
yi.chou@colorado.edu

**Eric Goubault**
LIX, CNRS, Ecole Polytechnique, IP-Paris, France
goubault@lix.polytechnique.fr

**Sylvie Putot**
LIX, CNRS, Ecole Polytechnique, IP-Paris, France
putot@lix.polytechnique.fr

## Abstract

In this paper, we propose polynomial forms to represent distributions of state variables over time for discrete-time stochastic dynamical systems. This problem arises in a variety of applications in areas ranging from biology to robotics. Our approach allows us to rigorously represent the probability distribution of state variables over time, and provide guaranteed bounds on the expectations, moments and probabilities of tail events involving the state variables. First, we recall ideas from interval arithmetic, and use them to rigorously represent the state variables at time $t$ as a function of the initial state variables and *noise symbols* that model the random exogenous inputs encountered before time $t$. Next, we show how concentration of measure inequalities can be employed to prove rigorous bounds on the tail probabilities of these state variables. We demonstrate interesting applications that demonstrate how our approach can be useful in some situations to establish mathematically guaranteed bounds that are of a different nature from those obtained through simulations with pseudo-random numbers.

## 1 Introduction

In this paper, we consider the problem of rigorously quantifying the uncertainty in the states of a stochastic dynamical system due to the influence of initial state uncertainties and stochastic disturbance inputs. This problem is fundamental to many applications. For instance, imagine a stochastic model of an aircraft whose motion is subject to unknown future wind forces. We wish to understand the distribution of the aircraft's position at some future time point of interest in order to make critical decisions whether the aircraft will collide with a building. Solving such problems requires us to know facts about the distribution of key state variables at some time instants $t$ in the future. In turn, we wish to use distributions to reason about quantities such as expectations and moments, as well as probabilities of events.

Our approach in this paper combines ideas from higher order interval arithmetic [19] and concentration of measure inequalities [11] to arrive at a framework to reason about uncertainties. (a) We represent state variables as multivariate polynomial forms over so-called noise symbols that represent some random variables with known, simple distributions. We use intervals that capture uncertainties that

can be arbitrarily correlated with these noise symbols. In doing so, we directly extend previous work by Bouissou et al, which considered affine (linear + constant) functions [5]. (b) We define basic arithmetic operations such as additions and multiplications over polynomial forms, in addition to more complex operations such as trigonometric functions and rational functions over polynomial forms. Finally, (c) we present approaches that can naturally provide upper bounds for *tail probabilities* such as $\mathbb{P}(X \geq t)$, for a random variable $X$ and a number $t$, using concentration of measure inequalities [11]. We demonstrate that the combination of ideas in this paper can be used to reason about interesting nonlinear and stochastic dynamical systems from the literature, providing direct comparisons with the related work of Bouissou et al (ibid).

Our comparison reveals many interesting aspects of our approach. Although it is potentially more expensive than the affine form approach, the approach in this paper provides tighter bounds in most cases when compared to affine forms. Furthermore, the polynomial form approach of this paper does not track correlations between the individual terms, in contrast to Bouissou et al, wherein a lot of effort is spent tracking such correlations over the terms of the affine form. This results in a computational advantage despite tracking higher order information. We also use the examples to show some of the challenging aspects of uncertainty propagation. In summary, we show that the approach presented here can be promising provided the degree bounds on the polynomials are carefully tuned to the problem at hand.

## 1.1 Related Work

The problem of approximating the non-linear image of distributions is of considerable practical importance, and has received attention from many different communities.

For the case of set-valued uncertainties, so-called *Taylor models*, originally proposed by Berz and Makino, use polynomials plus error intervals to represent the possible reachable states at some time $t$ [19, 21]. This been used for outer (and inner) approximating the image of non-linear functions, and for discrete, continuous [19, 21] and hybrid dynamical systems [7]. In the probabilistic case, similar methods [12, 26] provide polynomial approximations of non-linear images of probability distributions in the sense that any expectations and moments using expansions similar to Taylor models. These approaches guarantee weak convergence for probability distributions, as the degree of these polynomials is increased.

In this work, we similarly define polynomial approximations of non-linear images of distributions, based on simple polynomial arithmetics. The resulting representation, called polynomial forms, provide a stronger guarantee of *measure-theoretic* inclusion. In other words, polynomial forms represent a set of distributions, and furthermore, the actual image distribution is included in this set. In that sense, our work is more akin to the work on imprecise probabilities, such as Dempster-Shafer or P-box approximations [13, 25], but without the burden of potential combinatorial explosion of the size of the representation (e.g. number of focal elements), as has been observed in some of the related work [1, 5]. In the present work, instead of relying on interval arithmetic [20] and on affine forms [9] as in our previous work, we rely on polynomial forms that share some similarities with polynomial zonotopes [16], a generalization of zonotopes and Taylor models.

In uncertainty propagation work and surrogate models theory, similar ideas about polynomial expansions have been exploited. Originally, Wiener expansion was used to model stochastic processes with Gaussian random variables. In that case, an expansion can be made in terms of Hermite polynomials, a family of orthonormal polynomials. This has been later expanded to various distributions, still using a decomposition on a basis of orthonormal polynomials. This approach is known as (generalized) polynomial chaos, see e.g. [2]. Still, the method is known to be limited to a small number of random variables. In this work, we use a more classical expansion on a monomial basis, without the need to calculate suitable orthonormal representations, which are expensive to compute.

Our work also uses general concentration of measures inequalities [11] to derive bounds on tail probabilities that are of interest to risk analysis and safety verification. Concentration of measure inequalities have been applied in engineering, mostly for model validation against experimental data [14], but also for bounding the probability of failure events, see e.g. [18, 28]. In comparison, our work utilizes rigorously derived error bounds that transfer over to bounds that are rigorous, regardless of the degree bound on the polynomial form approximation. The error bounds also allow the user to

make trade-offs between faster computation achieved using lower degree polynomials and the quality of the bounds obtained knowing the associated error.

## 2 Preliminaries

Let $\mathbf{x} \in \mathbb{R}^n$ represent a $n \times 1$ vector of state variables. We will use $\mathcal{U}(a,b)$ to denote a random variable with a uniform distribution in the range $[a,b]$, $\mathcal{N}(\mu, \sigma)$ to denote a Gaussian random variable with mean $\mu$ and standard deviation $\sigma$, and $\mathsf{TruncNormal}(\mu, \sigma, [a,b])$ to denote a truncated normal distribution with mean $\mu$, standard deviation $\sigma$ and interval $[a,b]$.

**Definition 1** (Discrete-Time System). *A discrete time dynamical system $\Pi$ is defined by the tuple $\langle \mathbf{x}, f, \mathcal{W}, \mathcal{X}_0 \rangle$, wherein $\mathbf{x}$ represents the state variables that are updated at each time step as:*

$$\mathbf{x}(t+1) \ := \ f(\mathbf{x}(t), \mathbf{w}(t)), \ \mathbf{w}(t) \sim \mathcal{W},$$

*wherein $f : \mathbb{R}^{n+m} \to \mathbb{R}^n$ is a continuous function that maps the current state $\mathbf{x}(t)$ and current sample stochastic disturbances $\mathbf{w}(t)$ sampled from a probability distribution $\mathcal{W}$ to yield the next state $\mathbf{x}(t+1)$. The initial states of the system are sampled from an initial distribution $\mathbf{x}(0) \sim \mathcal{X}_0$.*

**Example 1** (Turning Vehicle Model). *Consider a model of vehicle with state variables $\mathbf{x} : (x, y, v, \psi)$ modeling the position $(x, y)$, velocity $v$, and yaw angle $(\psi)$. The vehicle's velocity is stabilized around $v_0 = 10$ m/s using a proportional feedback and its yaw is stabilized around $\psi_0 = 0.1$ radians. The dynamical equations are given by:*

$$x' \ = \ x + \tau v \cos(\psi), \ y' \ = \ y + \tau v \sin(\psi), \ v' \ = \ v + \tau(K_v(v - v_0) + w_1), \ and \ \psi' \ = \ \psi + w_2.$$

*Here, $x', y', v', \psi'$ denote the values of the state variables at time $t + 1$ and $x, y, \ldots, \psi$ denote state at time $t$. The disturbance inputs $w_1$ and $w_2$ are specified by their distributions: $w_1 \sim \mathcal{U}(-0.1, 0.1)$, $w_2 \sim \mathsf{TruncNormal}(0, 0.1, [-1, 1])$. Also, the constant parameter is set to $K_v = -0.5$. The initial values of the state variables are chosen according to some initial distributions: $x(0) \sim \mathcal{U}(-0.1, 0.1)$, $y(0) \sim \mathcal{U}(-0.5, -0.3)$, $v(0) \sim \mathcal{U}(6.5, 8.0)$, $\psi(0) \sim \mathsf{TruncNormal}(0, 0.1, [-1, 1])$.*

Given a system $\Pi$, we obtain sample trajectories of the system as a sequence of states $\mathbf{x}_i(0), \ldots, \mathbf{x}_i(t), \ldots$, wherein $\mathbf{x}_i(0)$ is a sample from the distribution $\mathcal{X}_0$ and $\mathbf{x}_i(t+1)$ is obtained from $\mathbf{x}_i(t)$ by (a) sampling $\mathbf{w}_i(t)$ according to $\mathcal{W}$ and (b) computing $\mathbf{x}_i(t+1) = f(\mathbf{x}_i(t), \mathbf{w}_i(t))$. Note that the individual disturbance inputs are all drawn independently of each other. However, each component of $\mathbf{x}(t)$ can be correlated with others due to the dynamical update.

The behavior of $\Pi$ can also be understood in terms of the distribution of states $\mathcal{X}_t$ at time $t$ with $\mathcal{X}_0$ specified as part of the system's description. Let $\mathbb{P}_t(S)$ denote the probability that $\mathbf{x}(t) \in S$ for a measurable set $S$, and furthermore, let $P_w$ denote the distribution function for the disturbance inputs $\mathbf{w}$. For a measurable set $S \subseteq \mathbb{R}^n$, we have

$$\mathbb{P}_{t+1}(S) \ = \ \int_{\mathbf{x}} \int_{\mathbf{w}} \mathbb{I}\left\{f(\mathbf{x}, \mathbf{w}) \in S\right\} P_w(d\mathbf{w}) \mathbb{P}_t(d\mathbf{x}).$$

Here $\mathbb{I}\{A\}$ denotes the indicator function for an event $A$. The continuity of $f(\mathbf{x}, \mathbf{w})$ guarantees that the integral on the right is well defined. Note that although $\mathcal{X}_0$, the initial probability distribution and $\mathcal{W}$, the distribution of

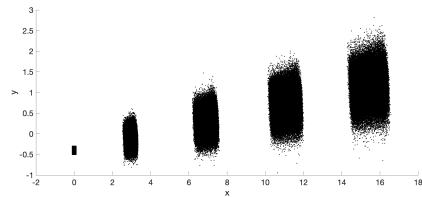

Figure 1: Samples from distributions $\mathcal{X}_0, \mathcal{X}_5, \mathcal{X}_{10}, \mathcal{X}_{15}$ and $\mathcal{X}_{20}$ for model from Example 1.

disturbances are known distributions from families such as Gaussian, Uniform or Exponential, the distribution $\mathcal{X}_t$ for $t > 0$ is often hard to reason about.

Figure 1 shows the distribution of trajectories at various time instants for the model from example 1. We support two types of queries involving the intermediate distributions $\mathcal{X}_t$, for some specified time $t \geq 0$, in this paper:

1. *Expectations/Moments:* Find bounds on $\mathbb{E}(h(\mathbf{x}(t)))$ for some function $h : \mathbb{R}^n \to \mathbb{R}$ over the state variables at some time $t$.

2. *Tail Probability Queries:* Find bounds on $\mathbb{P}(h(\mathbf{x}(t)) \geq c)$ for some function $h : \mathbb{R}^n \to \mathbb{R}$ over the state variables at some time $t$ and bound $c$. Note that lower tail bounds $\mathbb{P}(h(\mathbf{x}(t)) \leq c)$ can be rewritten as $\mathbb{P}(-h(\mathbf{x}(t)) \geq -c)$.

Going back to Ex. 1, we wish to know what the expectation of $x(10)$ and $y(10)$ are. We may also seek to know the probability that $x(10) \geq 8$.

## 3  Polynomial Forms

In this section, we present polynomial forms over noise symbols, to represent probability distributions $\mathcal{X}_t$ over the states $\mathbf{x}(t)$ at times $t \geq 0$. A key distinction, especially from other polynomial expansion approaches [22], is that we seek to capture $X_t$ conservatively by *bounding the error using intervals*.

First, we will define the noise symbols. A noise symbol denoted $w$ is a random variable with a known "primitive" distribution. An environment $\eta$ is a map from each noise symbol $w$ to its probability distribution $\eta(w)$. Let $W$ denote the set of noise symbols. We will assume that the noise symbols are all mutually independent.

A power product over variables $\mathbf{w} : (w_1, \ldots, w_k)$ is of the form $w_1^{r_1} \times w_2^{r_2} \cdots \times w_k^{r_k}$, for natural numbers $r_i \in \mathbb{N}$. For convenience, we write this power product as $\mathbf{w}^{\mathbf{r}}$ for vector $\mathbf{r} : (r_1, \ldots, r_k)$. The degree of the power product is given by $\sum_{j=1}^{k} r_j = ||\mathbf{r}||_1$. A multivariate polynomial over variables $\mathbf{w}$ is of the form $p(\mathbf{w}) : \sum_{i=0}^{m} a_i \mathbf{w}^{\mathbf{r}_i}$. The degree of the polynomial is the maximum degree amongst all the power products with nonzero coefficients.

**Definition 2** (Polynomial Forms). *A polynomial form $(p + I)$ over a set of noise symbols $N$ : $\{w_1, \ldots, w_k\}$ consists of a multivariate polynomial $p(w_1, \ldots, w_k)$ over the noise symbols and an "error" interval $I : [\ell, u]$.*

Formally, the *semantics* of a polynomial form is a set of functions over $\mathbf{w}$.

$$[\![p + I]\!] = \left\{ p(\mathbf{w}) + \hat{f}(\mathbf{w}) \ \middle| \ \hat{f}(\mathbf{w}) \text{ is measurable, and } (\forall \mathbf{w}) \ \hat{f}(\mathbf{w}) \in I \right\} .$$

Informally, a polynomial form represents two types of uncertainties:

(a) A *stochastic* component $p$ that is simply a polynomial function of the random variables represented by the noise symbols in the set $N$. A sample for this component is given by sampling the individual noise symbols from their distributions and then evaluating $p$.

(b) A *error* component $I$ that is assumed to be modeled by some *measurable function* $\hat{f} : (w_1, \ldots, w_k) \mapsto I$ that maps sampled values of the noise symbols to the interval $I$. The form of this function $\hat{f}$ is ignored and its range is simply retained. However, note that the choice in this interval can be arbitrarily correlated with that of $w_1, \ldots, w_k$.

(c) A sample from a polynomial form is thus taken to be the sum of a sample each from the stochastic and error components.

A polynomial form $(p + I)$ can also been seen as defining a *family* of possible distributions (measures) over $\mathbb{R}$. Let $[a, b)$ be an interval over the real line. A measure $\mu$ over the real numbers is compatible with the polynomial form $(p, I)$ iff for every interval $[a, b)$ over the reals, $\mu([a, b)) \in \{\mathbb{P}(p(\mathbf{w}) + x \in [a, b)) \mid x \in I\}$. The probability $\mathbb{P}(\cdot)$ in the definition above is computed over the joint distribution of the noise symbols in $W$.

**Example 2.** *Let $w_1$ denote the distribution $\mathcal{N}(0, 1)$ and $w_2$ denote $\mathcal{U}(-1, 1)$. The polynomial form $(w_1 + w_2^2) + [-0.1, 0.1]$ denotes a family of functions over $w_1, w_2$. For example, here is one member of this family:*

$$p_j : \begin{cases} w_1 + w_2^2 - 0.1 \sin(w_1 + w_2) & \text{if } w_1 \leq 0, \ w_2 \leq 0 \\ w_1 + w_2^2 & \text{if } w_1 \leq 0, w_2 \geq 1 \\ w_1 + w_2^2 + 0.1 \cos^2(w_2) & \text{otherwise} \end{cases} .$$

**Remark 1.** *We note that a polynomial form $p + I$ thus represents an unknown function of the form $p(\mathbf{w}) + \hat{f}(\mathbf{w})$, wherein $\hat{f}$ belongs to the interval $I$. We require $\hat{f}$ to be measurable. Where appropriate, we can justify further assumptions about $\hat{f}$, in terms of the existence of moments $\mathbb{E}(\hat{f}^k)$.*

## 3.1 Calculus of Polynomial Forms

We describe elementary operations over polynomial forms, including polynomial form arithmetic and the application of functions such as sine and cosine.

Let $\text{range}(p)$ for polynomial $p(\mathbf{w})$ denote the range of possible values of $p_i(\mathbf{w})$. The range can be computed using standard interval arithmetic approaches that treat each random variable as an interval ranging over its set of support. We assume basic interval operations such as $\oplus$ for adding two intervals, and $\otimes$ for multiplying intervals (Cf. [9, 19, 20]).

**Linear Combinations:** Given two polynomial forms $(p_1 + I_1)$ and $(p_2 + I_2)$, their sum is given by the form $p_1 + p_2 + (I_1 \oplus I_2)$. Likewise, the product of a form $p + I$ by a scalar $\lambda$ is given by the form $\lambda p + \lambda I$.

**Multiplication:** Let $p_1 + I_1$ and $p_2 + I_2$, be the two forms. The multiplication of the forms, denoted $(p_1 + I_1) \otimes (p_2 + I_2)$, is given by the form $p + I$ with polynomial component $p : p_1 \times p_2$ and the interval $I : (I_1 \otimes I_2) \oplus (I_1 \otimes \text{range}(p_2)) \oplus (I_2 \otimes \text{range}(p_1))$.

**Lemma 1** (Soundness of Multiplication). *For given polynomial forms $(p_1 + I_1)$ and $(p_2 + I_2)$* $[\![(p_1 + I_1) \otimes (p_2 + I_2)]\!] \supseteq \{f_1 \times f_2 \mid f_1 \in [\![p_1 + I_1]\!], \ f_2 \in [\![p_2 + I_2]\!] \ \}$.

Proofs are included in the appendix as part of the supplementary materials.

**Truncation:** Often polynomial forms grow in degree, especially due to operations such as multiplication. Therefore, we can truncate the polynomial by removing power products that exceed some maximum degree bounds. Formally, for polynomial form $p + I$: (a) Write $p = p_1 + p_2$ wherein $p_1$ has all those power products with degree less than the cutoff $K$, and $p_2$ has the power products with degree greater than $K$. (b) The form $\text{trunc}_D(p + I)$ is then defined as $p_1 + (I \oplus \text{range}(p_2))$.

**Continuous Functions:** Let $g : \mathbb{R} \to \mathbb{R}$ be a function that is continuous and $m$ times differentiable. Let $p + I$ be a polynomial form. We wish to compute a polynomial form corresponding to $g(p + I)$. To do so, we proceed as follows:

1. Choose a "center" point $c$ given by the midpoint of $\text{range}(p)$.
2. Perform a Taylor series expansion of $g$ around $c$ using the first $j + 1$ derivatives of $g$ where $j + 1 \leq m$: $g(c + h) = g(c) + g'(c)h + \cdots + g^{(j)}(c)\frac{h^j}{j!} + R_{j+1}$.
3. Substitute the polynomial form $(p - c + I)$ for $h$ using $\otimes$ as the multiplication operator.
4. The Lagrange remainder $R_{j+1}$ is given by estimating the range of the function: $g^{(j+1)}(x)\frac{x^{j+1}}{(j+1)!}$ for $x \in \text{range}(p + I)$.

Let $\hat{p} + \hat{I}$ be the result of carrying out the computations above for a polynomial form $p + I$.

**Lemma 2.** *For a $m$ times differentiable function $g$ and $j + 1 \leq m$, then $[\![g(p + I)]\!] \subseteq [\![\hat{p} + \hat{I}]\!]$.*

Using the idea above, we can compute functions such as $\sin$, $\cos$ and $\exp$ over polynomial forms.

## 3.2 Polynomial Form Uncertainty Propagation

Thus far, we have encountered polynomial forms and studied how to perform basic calculations such as arithmetic and function applications. We will now extend this to represent the distributions $\mathcal{X}_t$ at time $t$ for a dynamical system $\Pi$. Before we do so, we make some simplifying assumptions.

1. *Compactly supported distributions:* We will assume that all the distributions in $\Pi$, including the initial states and the distribution of the disturbances at each step, have a compact support. Whereas this is somewhat restrictive, it is possible to *artificially* truncate distributions such as Gaussian to a compact interval, and account for the probability of a sample falling outside of the truncation in our probability calculations (Cf. [24, Section 3.2]).
2. *All moments exist:* All the moments $\mathbb{E}(w^j)$ exist for $j \geq 1$.
3. *Smooth Updates:* We assume that the update function $f(\mathbf{x}, \mathbf{w})$ of the dynamical system is $C^\infty$: I.e, it is continuous and arbitrarily differentiable.

The approach begins by mapping each state variable $x_j(0)$ with a corresponding polynomial form $p_{j,0} + I_{j,0}$. Initial $p_{j,0} = w_{j,0}$ and $I_{j,0} = [0,0]$, wherein $w_{j,0}$ is a noise symbol with distribution given by the initial distribution of the state variable $x_j(0)$.

At the $t^{th}$ step, we obtain polynomial forms $p_{j,t} + I_{j,t}$ corresponding to $x_j(t)$. Since, $\mathbf{x}(t+1) = f(\mathbf{x}(t), \mathbf{w}(t))$, we introduce fresh noise symbols corresponding to $\mathbf{w}(t)$ and substitute each $\mathbf{x}_j(t)$ by the polynomial form $p_{j,t} + I_{j,t}$. Next, we use the polynomial form evaluation over the function $f$ to compute a polynomial form for $\mathbf{x}(t+1)$.

**Example 3.** *Consider again the example 1. For time $t = 0$, we initialize the state variables to polynomial forms, by introducing noise symbols $w_0, w_1, w_2$ and $w_3$. The state variables at time $t = 0$ are $x(0) : w_0$, $y(0) : w_1$, $v(0) : w_2$, $\psi(0) : w_3$, wherein $w_0 : \mathcal{U}(-0.1, 0.1)$, $w_1 : \mathcal{U}(-0.5, -0.3)$, $w_2 : \mathcal{U}(6.5, 8.0)$ and $w_3 : \mathsf{TruncNormal}(0, 0.1, [-1, 1])$. Note that $x(1) : x(0) + 0.1 v(0) \cos(\psi(0))$. Carrying out these calculations using the polynomial forms yields the form for $x(1)$.*

$$x(1) : \ w_0 + 0.0999306 w_2 - 0.05 w_2 w_3^2 + [-0.00055, 0.034].$$

*As we propagate the polynomial forms from one time step to another, we obtain larger polynomials involving increasing number of noise symbols. At time $t = 20$, the polynomial for $x(20)$ involves $42$ noise symbols and $269$ terms:*

$$x(20) : \ [7.13, 7.60] - 0.31 w_{41}^2 + 0.026 w_{41}^4 + \cdots - 0.045 w_2 w_7^2 - 0.048 w_2 w_5^2 - 0.05 w_2 w_3^2 + w_0.$$

### 3.3 Bounding Expectations and Moments

We will now describe how to bound expectations and moments of polynomial forms. Let $(p, I)$ be a polynomial form involving noise symbols $\mathbf{w}$ with the assumptions stated in Section 3.2. Let $\hat{f}(\mathbf{w})$ be a continuous function over noise symbols $\mathbf{w}$, wherein $\mathbf{w}$ belongs to a compact set, and furthermore $\hat{f}(\mathbf{w}) \in I$ for $l, u \in \mathbb{R}$. For any $j \in \mathbb{N}$, let $I^j$ denote the interval over approximation of the set $\{x^j \mid x \in I\}$.

**Lemma 3.** *For any $j \in \mathbb{N}$, the moment $\mathbb{E}(\hat{f}^j)$ is bounded by the interval $I^j$.*

Using this, we can bound the expectation of a polynomial form $p + I$. Let $p = \sum_j a_j \mathbf{w}^{\mathbf{r}_j}$. To compute $\mathbb{E}(p) : \sum_j a_j \mathbb{E}(\mathbf{w}^{\mathbf{r}_j})$. Each power-product $\mathbf{w}^{\mathbf{r}_j}$ is expanded out as $\prod_{i=1}^k w_i^{r_{j,i}}$. We use the pairwise independence of the noise symbols to compute its expectation as $\prod_{i=1}^k \mathbb{E}(w_i^{r_{j,i}})$. For each noise symbol $w_i$, the environment $\eta$ maps it to a distribution whose moment generating function is assumed to be known. The overall expectation of the form is bounded by the interval $\mathbb{E}(p) + I$.

A similar approach can be utilized for higher-order moments. However, they require us to first compute $(p + I)^j$ using the multiplication operation defined over polynomial forms. The subsequent section presents an approach to make the computation of higher moments more efficient.

## 4 Concentration of Measure

In this section, we study how to provide bounds on tail probabilities $\mathbb{P}(p + I \geq c)$ (equivalently $\mathbb{P}(p + I) \leq -c$). Let $I$ be the interval $[l, u]$. We note that $\mathbb{P}(p + I \geq c) \leq \mathbb{P}(p \geq c - u)$ since $p + [l, u] \geq c$ implies that the latter event $p \geq c - u$. For queries of the form $\mathbb{P}(h(p + I) \geq c)$, we first evaluate the function $h(p + I)$ as a polynomial form, reducing it to the former problem. Next, since we can compute $\mathbb{E}(p)$, we restate the problem as $\mathbb{P}(p - \mathbb{E}(p) \geq c - u - \mathbb{E}(p))$.

Computing or bounding the probability requires us to bound an integral over the noise symbols $\mathbf{w}$. These integrals involve indicator functions and are not known in a closed form. For instance, the polynomial form shown in Ex. 3 has $42$ noise symbols and $269$ terms, making the integral all but impossible to compute precisely. In what follows, we present concentration of measure inequalities to estimate upper bounds.

Let $s : c - u - \mathbb{E}(p)$. We will now recall various inequalities that can be used and provide approaches to calculate the bounds. A more in-depth presentation can be found elsewhere [11].

**Chebyshev Inequality:** $\mathbb{P}(p - \mathbb{E}(p) \geq s) \leq \frac{\mathbb{E}((p - \mathbb{E}(p))^2)}{s^2}$.

The numerator is the variance of $p$. To compute it, we can take the square of the polynomial $p - \mathbb{E}(p)$, and compute its expectation. Likewise, we may extend Chebyshev inequality to consider higher moments $2m$ for $m \geq 1$. $\mathbb{P}(p - \mathbb{E}(p) \geq s) \leq \frac{\mathbb{E}((p - \mathbb{E}(p))^{2m})}{s^{2m}}$. However, such a computation can be

expensive if $p$ is a large polynomial involving numerous terms. In such a case, we can exploit the structure of $p$ to split it into mutually independent components. We will discuss these simplifications further and show how they enable the application of other concentration inequalities.

## 4.1 Splitting Polynomial Forms

We briefly discuss ideas around making the polynomial forms sparser to minimize the computational overhead while maintaining the tightness of the bounds on the expectation and probability queries. In theory, if we restrict ourselves to polynomials of degree at most $d$, the number of terms in the polynomial form after $n$ steps grows as $O((n|W|)^d)$, wherein $|W|$ is the number of disturbance variables in the system and $d$ is the degree limit. This can be prohibitively expensive as $n$ grows in the thousands, even for a small degree $d$. We present ideas to control the growth in the size of the polynomial form and make the calculations for expectations and bounds tractable: (a) *Truncation*: moving terms with small ranges from the polynomial to the interval part to reduce the polynomial size while keeping the error bounds small. We have discussed truncation previously in Section 3.1. (b) *Splitting*: writing the polynomial form as the sum of mutually independent components in order to make the calculation of higher moments faster; and finally (c) *Partitioning*: balanced partition approach to select terms that can be truncated to improve our ability to split.

**Splitting Polynomials into Components:** We describe splitting of polynomials to help us apply Chebyshev bounds efficiently and also allow us to apply bounds Chernoff and Bernstein bounds. The idea is to split a polynomial form as a sum of mutually independent polynomials involving a subset of the variables. Let $p := \sum_{j=1}^{m} a_j \mathbf{w}^{\mathbf{r}_j}$ be a polynomial form. A noise symbol $w_i$ is related to $w_j$ iff there exists a term in $p$ that contains both $w_i$ and $w_j$ with positive powers.

**Example 4.** *Consider the polynomial $p = w_1 w_2 + w_1^2 + w_3^2 + w_4 w_5$. We note that the relation consists of the pairs $(w_1, w_2)$ and $(w_4, w_5)$.*

Naturally, the relation between noise symbols forms an undirected graph $G_p$ that connects two noise symbol by an edge if they occur in the same power-product. Next, we compute the strongly connected components of $G_p$. This defines a partition of the noise symbols $W$ into disjoint subsets $W_1, \ldots, W_j$, and a corresponding "splitting" of $p$ into $p_1, \ldots, p_j$ such that $p = p_1(\mathbf{w}_1) + \cdots + p_j(\mathbf{w}_j)$.

**Example 5.** *Going back to Ex. 4, we note that the set of noise symbols is partitioned into $\{w_1, w_2\}$, $\{w_3\}$ and $\{w_4, w_5\}$. Likewise, the polynomial $p$ is now split into $p = p_1(w_1, w_2) + p_2(w_3) + p_3(w_4, w_5)$ wherein $p_1 : w_1 w_2 + w_1^2$, $p_2 : w_3^2$ and $p_3 : w_4 w_5$.*

Splitting a polynomial into components can allow us to apply Chebyshev bounds more efficiently. Recall that applying Chebyshev bounds requires us to compute $\mathbb{E}((p - \mathbb{E}(p))^{2m})$. (a) The polynomials $p_1, \ldots, p_j$ are functions of mutually disjoint sets of random variables. Therefore, $p_i$ is independent from $p_j$. (b) The variance $V(p) : \mathbb{E}((p - \mathbb{E}(p))^2)$ can be decomposed as $\sum_{i=1}^{j} V(p_i)$. (c) The fourth moment $\mathbb{E}((p - \mathbb{E}(p))^4)$ can be decomposed as $\sum_{i=1}^{j} \mathbb{E}((p_j - \mathbb{E}(p_j))^4) + 2 \sum_{i=1}^{j-1} \sum_{l=i+1}^{j} V(p_i) V(p_l)$. Similarly, we can obtain computationally efficient means to calculate the higher order central moments of $p$ in terms of $p_i$.

Thus the decomposition of a large polynomial into mutually independent parts allows us to save time when computing the central moments. Splitting a polynomial $p$ into pairwise independent components also allows us to utilize inequalities over sums of random variables such as Chernoff-Hoeffding and Bernstein bounds.

**Chernoff-Hoeffding Bounds:** Let range$(p_i) : [a_i, b_i]$. The Chernoff-Hoeffding bounds are as follows [15]: $\mathbb{P}(p - \mathbb{E}(p) \geq s) \leq \exp\left(-\frac{2s^2}{\sum_{i=1}^{j}(b_i - a_i)^2}\right)$.

**Bernstein Bounds:** Using the $V(p_i)$ of the individual components $p_i$, yield Bernstein inequalities [4]. Let $M$ be a number chosen such that $|p_i - \mathbb{E}(p_i)| \leq M$ for each $i = 1, \ldots, j$. We have $\mathbb{P}(p - \mathbb{E}(p) \geq s) \leq \exp\left(\frac{-s^2}{2\sum_{i=1}^{j} V(p_i) + \frac{2sM}{3}}\right)$.

**Example 6.** *Consider again the turning vehicle model from Ex. 1. Our goal is to bound the probability $\mathbb{P}(y(8) \geq 2)$. The polynomial form for $y(8)$ is computed with truncation applied to terms of degree 7 or more, at each step of the computation. The result is a polynomial form over 20 noise symbols*

*of degree* 6. *Using Chebyshev bounds, we conclude that* $\mathbb{P}(y(8) \geq 2) \leq 0.009$. *However, Chernoff inequality yields a bound* $0.95$ *whereas Bernstein inequality yields* $0.81$.

In general, for a problem of the form $\mathbb{P}(p - \mathbb{E}(p) \geq s)$, Chebyshev bounds are of the form $\beta s^{-2}$ whereas Chernoff bounds are of the form $\exp(-\gamma s^2)$. As $s$ grows larger, the latter bounds will be tighter. The constants $\beta, \gamma$ are computed by our approach.

**Balanced Partitioning of Polynomials:** A final optimization enables us to selectively move terms from the polynomials to the error interval in order to enable splitting a large polynomial involving many noise symbol into the sum of smaller mutually independent polynomials, while at the same time keeping the overall width of the error interval within bounds. We show that this process can be reduced to a mixed integer optimization problem to perform a "balanced partitioning" of a weighted graph [3], whose vertices represent noise symbols and weighted edges represent noise symbols that occur together with the weight representing how much the width of the error interval. We will describe this approach and a preliminary study in our supplementary material. A detailed study will be provided in an extended version of this paper.

# 5 Experiments

In this section, we describe an evaluation of our approach using some challenging nonlinear stochastic systems taken from the literature. We present a comparison of our work with the related approach of Bouissou et al [5], which uses affine forms. Although affine forms are degree one polynomials, Bouissou et al approach also includes the creation of new noise symbols representing nonlinear functions of previous noise symbols along with correlations between them.

Our evaluation is based on a C++-based prototype implementation that reads in the description of a nonlinear dynamical system over a set of system and disturbance variables. The dynamics can currently include polynomials, rational and trigonometric functions. Our implementation uses the MPFI library to perform interval arithmetic in order to rigorously bound the errors and guarantee soundness [23]. On the side, we manually implemented each benchmark as a python program in order to perform numerical simulations using pseudorandom numbers for comparison.

Table 1 shows the results over a set of 8 benchmark problems. taken from various domains such as robotics, physics and biology. Each benchmark involves a discrete-time nonlinear stochastic model along with some "queries" that take the form of *computing expectations* of state variables, or *bounding probabilities* of simple properties. A detailed description of each benchmark and properties are available as part of the appendix.

First, the results clearly demonstrate that polynomial forms are more computationally expensive in many cases but nevertheless, can provide tighter bounds when compared to affine forms. Curiously, the affine form approach times out for the Cartpole model which involves repeated multiplications of affine forms followed by $\sin / \cos$ operations. This causes the size of the forms to double at each step. Polynomial forms avoid this blowup by maintaining higher order terms involving a smaller set of noise symbols in this case.

Next, we note that in almost all cases, the polynomial form approach is able to provide very tight interval bounds on the expectations. Note that these bounds are rigorous due to the rigorous interval arithmetic used in our approach.

Finally, we compare the bounds placed on probabilities. Herein, the polynomial form drastically improves upon the affine form approach. Nevertheless, the conservative nature of concentration of measure inequalities is clearly seen. In many cases, however, our approach places upper bounds on the probabilities of the rare events that are not seen in simulations. Such bounds help us prove guarantees for these stochastic systems.

# 6 Conclusion

To conclude, we have provided an approach that provides guaranteed bounds on the expectations of state variables for nonlinear stochastic dynamics, and show how concentration inequalities can provide bounds for probabilities of tail inequalities. In the future, we propose to investigate further

Table 1: Results on nonlinear benchmark models comparing our approach (Poly. Form) against the affine form approach [5] and $10^5$ simulations. $|X|, |W|$: # state and disturbance variables, $N$: # time steps, $D$: degree of polynomial form. All timings indicated by the ⏱ symbol are in seconds. A detailed description of the models is provided in the appendix.

| Benchmark | $(|X|, |W|)$ | $N$ | $D$ | Poly. Form | Aff. Form [5] | $10^5$ Sim. |
|---|---|---|---|---|---|---|
| Rimless | (1,1) | 2000 | 2 | ⏱46.1 s | ⏱75s | ⏱341.3s |
| Wheel [27] | | | $\mathbb{E}(x)$ | [1.791,1.792] | [1.02,3.73] | 1.7911 |
| | | | $\mathbb{P}(x \leq 0)$ | 0.036 | 0.61 | 0 |
| 2DRobot | (3,2) | 100 | 2 | ⏱35.8s | ⏱30.1s | ⏱143.4s |
| Arm [5] | | | $\mathbb{E}(x)$: | [268.87,268.88] | [268.6,270.7] | 268.85 |
| | | | $\mathbb{P}(x \geq 272)$: | 7E-8 | 1E-2 | 0 |
| Cartpole | (4,1) | 8 | 2 | ⏱45s | | ⏱3.5s |
| [27] | | | $\mathbb{E}(x)$: | [-0.176,0.196] | | 0.0076 |
| | | | $\mathbb{E}(\theta)$ : | [-0.344,0.346] | ⏱Timeout | 0.0014 |
| | | | $\mathbb{P}(x \geq 2)$ : | 0.006 | > 1 hour | 0 |
| | | | $\mathbb{P}(\theta \geq \pi/6)$ : | 0.017 | | 0 |
| Ebola | (5,0) | 25 | 6 | ⏱10.1s | ⏱1.5s | ⏱1.3s |
| [8] | | | $\mathbb{E}(I)$: | [0.0983,0.09834] | [0.0283,0.157] | 0.098 |
| | | | $\mathbb{E}(e)$: | [0.0756,0.0757] | [-0.058,0.184] | 0.0757 |
| | | | $\mathbb{P}(e \geq 0.1)$: | 0.037 | 1 | 3.2E-4 |
| | | | $\mathbb{P}(I \leq 0.05)$: | 0.03 | 1 | 0 |
| Honeybee | (5,0) | 25 | 4 | ⏱52.9s | ⏱13.6s | ⏱28.5s |
| [6, 10] | | | $\mathbb{E}(z_1)$: | [241.099,241.103] | [16.1887,512.898] | 241.101 |
| | | | $\mathbb{E}(z_2)$: | [85.393,85.3952] | [-49.5456,238.419] | 85.396 |
| | | | $\mathbb{P}(z_1 \geq 265)$: | 0.018 | 1 | 0 |
| | | | $\mathbb{P}(z_2 \leq 60)$: | 0.014 | 1 | 0 |
| Coupled | (6,3) | 15 | 2 | ⏱60.8s | ⏱1.1s | ⏱2.3s |
| Vanderpol | | | $\mathbb{E}(y_3)$: | [0.427,0.432] | [0.409,0.451] | 0.429 |
| Oscillator | | | $\mathbb{P}(y_3 \geq 0.6)$ | 0.21 | 0.89 | 7.21E-4 |
| | | | $\mathbb{P}(y_3 \leq 0.2)$ | 0.12 | 0.74 | 0 |
| Laub-Loomis | (7,0) | 25 | 4 | ⏱253.9s | ⏱8.2s | ⏱1.97s |
| Network | | | $\mathbb{E}(x_1)$: | [0.9429,0.9436] | $[-\infty, \infty]$ | 0.9433 |
| [17] | | | $\mathbb{E}(x_2)$: | [0.7592,0.7624] | $[-\infty, \infty]$ | 0.761 |
| | | | $\mathbb{P}(x_1 \leq 0.7)$: | 0.0029 | 1 | 0 |
| | | | $\mathbb{P}(x_1 \geq 0.95)$: | 0.0145 | 1 | 0 |
| 10x1 Lattice | (10,0) | 15 | 4 | ⏱71.8s | ⏱63.9s | ⏱3.6s |
| Particles | | | $\mathbb{E}(u_1)$: | [-0.0523,-0.0521] | [-0.36,-0.01] | -0.0521 |
| [29] | | | $\mathbb{E}(u_8)$: | [0.0412,0.0425] | [-0.072, 0.068] | 0.0418 |
| | | | $\mathbb{P}(u_1 \geq 0)$: | 5E-4 | 0.99 | 0 |
| | | | $\mathbb{P}(u_8 \leq 0)$: | 0.81 | 1 | 0.1385 |

applications of concentration of inequalities such as the method of bounded differences/variances to establish bounds on how each disturbance input affects the final property of interest [11].

## 7  Broader Impact

We, as a society, rely on mathematical models and their predictions a lot more than we realize. The current COVID-19 pandemic, the kerfuffle during winter 2019 over the predicted course of hurricane Dorian, or one of the many causes of the 2008 great depression involving trading in derivatives based on faulty modeling assumptions, all bear witness to this fact. Beyond this, many safety critical applications to autonomous control systems such as closed loop medical devices rely on selecting optimal control strategies based on predictions provided by dynamical system models.

The present work uses tail inequalities to mechanize the derivation of mathematically rigorous bounds on predictions over future states of stochastic dynamical models. This has beneficial impacts in that, where it is applicable, such analysis can help us carefully analyze models and quantify uncertainty in the model predictions. For instance, our work may be used to predict that an aircraft flown using the control strategy in Ex. 1 will remain free from collision with building 30 meters away from its predicted trajectory with at least $99\%$ confidence.

At the same time, it is important to realize that in many instances, assumptions that are made such as "this disturbance behaves like a Gaussian random variable" are, at best, approximations. These approximations often become less valid when higher order moments or the tail behavior of these random variables are examined. We note that this paper uses approaches that rely intimately on knowing the higher order moments and reasoning about tail behaviors of polynomial functions of random variables. Negative impacts could include a false sense of confidence or trust in the reliability of a prediction in the case of forecast models or in a control strategy, derived from modeling assumptions that may be unsound. This negative impact is best mitigated in multiple ways which we hope to address in our future work: (a) provide a sensitivity analysis on how the conclusions change if the distributions are perturbed; or (b) work with partially specified families of distributions with uncertain expectations, variances and higher moments. We note that our approaches in this paper are highly suited for the latter approach.

## Acknowledgments and Disclosure of Funding

We thank the anonymous reviewers for helpful comments.

SS and CY acknowledge support from the US National Science Foundation under award numbers CPS 1836900, CCF 1815983; and the US Air Force Research Laboratory (AFRL).

EG and SP acknowledge support from the academic chair "Complex Systems Engineering", Ecole Polytechnique-ENSTA Paris-Télécom Paris-Thalès-Dassault Aviation-Naval Group-DGA-Fondation ParisTech-FX and AID project "Validation of Autonomous Drones and Swarms of Drones".

The authors do not have any outside competing interests to disclose.

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
