[Supplementary Material]

# A  Details of the Benchmark Models

**Rimless Wheel Walker**

This system describes a model of human walking called the rimless wheel model taken from the description provided by Steinhardt et al [27]. The rimless wheel system is described by a single state variable $x$ with the update equation:

$$x(t+1) \quad = \quad \cos^2(\theta)\left(x(t) + \tfrac{2g}{L}(1 - \cos(\beta_1))\right) - \tfrac{2g}{L}(1 - \cos(\beta_2)).$$

Here $x(t)$ models the squared angular velocity of the wheel rolling down an inclined plane between the two impacts of its spokes. The angle $\theta = 30$ degrees represents the angle between the wheel spokes. We define $\beta_1 : \frac{\theta}{2} + w(t)$ and $\beta_2 : \frac{\theta}{2} - w(t)$, wherein $w(t)$ is a truncated normal random variable with mean $\gamma_0 = 4$ degrees, representing the angle of the inclined plane, and standard deviation 1.5 degrees. The range of the variable is set to $[\gamma_0 - 9, \gamma_0 + 9]$. The spoke length $L$ is taken to be 1 meter and $g$ is taken to be $10m/s^2$. Following Steinhardt et al (ibid), our goal is to explore the distribution of $x(t)$ for various values of time $t$ and different values of $\gamma_0$. In particular, we are interested in estimating bounds on the probability $\mathbb{P}(x(t) \leq 0)$, which according to the model indicates that the wheel stops turning. Note that although $x(t)$ is supposed to model a non-negative quantity, nothing stops it from going negative in the model itself.

**Robotic Arm**

This system models the position of a 2D robotic arm end effector subject to various repeated translational and rotational movements, but incurring an error at each step modeled as a stochastic disturbance. A detailed description is available from Bouissou et al [5] and Sankaranarayanan [24], and expressed in the form of a Python program below.

```
1  angles = [10, 60, 110, 160, 140, ...
2           100, 60, 20, 10, 0]
3  x := TruncGaussian(0,0.05,-0.5,0.5)
4  y := TruncGaussian(0, 0.1,-0.5,0.5)
5  # run for 100 repetitions
6  for reps in range(0,100):
7  #iterate through angles
8    for theta in angles:
9      # Distance travelled variation
10     d = Uniform(0.98,1.02)
11     # Steering angle variation
12     t = deg2rad(theta) * (1 + ...
13         TruncGaussian(0,0.01,-0.05,0.05))
14     # Move distance d with angle t
15     x = x + d * cos(t)
16     y = y + d * sin(t)
17 #Probability that we went too far?
18 assert(x >= 277)
```

As the program illustrates, the main state variables are $(x, y)$ which are updated at each step as $x' = x + d\cos(\theta_i)$ and $y' = y + d\sin(\theta_i)$ for various angles $\theta_i$ in the array `angles` (see line 2). At each iteration, for each angle in the array, we draw a value of $d$ uniformly from the range $[0.98, 1.02]$ (see line 10) and the angle itself is subject to some error (see line 13).

**Cartpole Model**

The cartpole model is taken from the description provided by Steinhardt and Tedrake [27]. The overall system is the discretization of a ODE with a fixed time step $\delta = 0.1$. The state variables involved are $(x, \dot{x}, \theta, \dot{\theta})$. The update equations are:

$$
\begin{aligned}
x' &= x + \delta\dot{x} + 0.03w_1 \\
\theta' &= \theta + \delta\dot{\theta} + 0.03w_2 \\
\dot{x}' &= \dot{x} + \delta(-0.75\theta^3 - 0.01\theta^2 u - 0.05\theta\dot{\theta}^2 + 0.98\theta + 0.1u) + 0.1w_3 \\
\dot{\theta}' &= -5.75\theta^3 + \delta(-0.12\theta^2 u - 0.10\dot{\theta}^2 + 21.56\theta + 0.2u) + 0.1w_4
\end{aligned}
$$

Here $u$ refers to the feedback term that is computed as

$$u = -10x + 289.93\theta - 19.53\dot{x} + 63.25\dot{\theta}.$$

Likewise, $w_1, \ldots, w_4$ are truncated Gaussian RVs with range $[-1, 1]$, mean 0 and standard deviation given by $\sqrt{\delta}$.

The goal of the system is to stabilize the cart to a "neutral position" with $x = \dot{x} = \theta = \dot{\theta} = 0$. We ask for the probability that $x \geq 2$ and $\theta \geq \frac{\pi}{6}$.

**Ebola Model:** The Ebola infection model is an adaptation of the SIR model for infection transmission whose parameters have been identified based on an Ebola epidemic in Congo. The model has 5 state variables $(s, e, i, r, c)$ and the initial distributions are given by

$$s \sim \mathsf{TruncNormal}(0.7, 0.02, [0.6, 0.8]), \; e \sim \mathcal{U}(0.2, 0.4), \; i, r, c \sim \mathcal{U}(0, 0.04)$$

The dynamics are given by

$$
\begin{aligned}
s' &= s - (0.35si)0.5 \\
e' &= e + ((0.35si) - (0.28)e)0.5 \\
i' &= i + (0.28e - 0.29i)0.5 \\
r' &= r + (0.29i)0.5 \\
c' &= c + 0.28e0.5
\end{aligned}
$$

We are interested in the expectations of $i, e$ and the probabilities for the events $e \geq 0.1$ and $i \leq 0.05$.

**Honeybee Model:** This model is originally from the work of Britton et al [6] with parameters given by Dreossi et al [10]. The model captures how honeybees decide between two different nesting sites. The state variables include $(x, y_1, y_2, z_1, z_2)$. The initial distributions are

$$x \sim \mathsf{TruncNormal}(475, 5, [450, 500]), \; y_1 \sim \mathcal{U}(350, 400), \; y_2 \sim \mathcal{U}(100, 150)$$
$$z_1, z_2 \sim \mathsf{TruncNormal}(35, 1.5, [20, 50])$$

The dynamics are given by

$$
\begin{aligned}
x' &= x + 0.1(-0.001xy_1 - 0.001xy_2) \\
y_1' &= y_1 + 0.1(0.001xy_1 - 0.3y_1 + 0.0005y_1z_1 + 0.0007y_1z_2) \\
y_2' &= y_2 + 0.1(0.001xy_2 - 0.3y_2 + 0.0005y_2z_2 + 0.0007y_2z_1) \\
z_1' &= z_1 + 0.1(0.3y_1 - 0.0005y_1z_1 - 0.0007y_2z_1) \\
z_2' &= z_2 + 0.1(0.3y_2 - 0.0005y_2z_2 - 0.0007y_1z_2)
\end{aligned}
$$

The properties include the expectation of $z_2$ after 25 time steps, the probabilities that $z_1 \geq 265$ and $z_2 \leq 60$ all after 25 time steps.

**Coupled Vanderpol Oscillator:** A Vanderpol oscillator is a two dimensional nonlinear dynamical system model that exhibits a stable limit cycle. We study a model of three coupled oscillators with a simple linear topology. The state variables are $(x_1, y_1, x_2, y_2, x_3, y_3)$ wherein $(x_i, y_i)$ represent the states of the $i^{th}$ oscillator.

$$x_1 \sim \mathcal{U}(1.0, 1.2), y_1 \sim uniform(0.85, 0.95), x_2 \sim \mathcal{U}(1.3, 1.4), y_2 \sim \mathcal{U}(2.1, 2.3)$$
$$x_3 \sim \mathcal{U}(-0.1, 0.1), \; y_3 \sim \mathcal{U}(0.3, 0.5)$$

The dynamics are given by:

$$
\begin{aligned}
x_1' &= x_1 + 0.05y_1 + w_1 \\
y_1' &= y_1 + 0.05(0.5(1.0 - x_1^2)y_1 - x_1 + 0.05x_2) \\
x_2' &= x_2 + 0.05y_2 + w_2 \\
y_2' &= y_2 + 0.05(0.33(1.0 - x_2^2)y_2 - x_2 + 0.05x_3) \\
x_3' &= x_3 + 0.05y_3 + w_3 \\
y_3' &= y_3 + 0.05(0.45(1.0 - x_3^2)y_3 - x_3)
\end{aligned}
$$

$w_1, w_2, w_3$ are random disturbances drawn from $\mathcal{U}(-0.01, 0.01)$.

**Laub-Loomis Biological Network:** The Laub-Loomis model taken from [17] is a biochemical reaction network that exhibits oscillatory properties. The state variables are $(x_1, \ldots, x_7)$ with the following initialization:

$$
\begin{aligned}
x_1 &\sim \mathcal{U}(1.0, 1.2) \\
x_2 &\sim \mathcal{U}(0.85, 1.05) \\
x_3 &\sim \mathcal{U}(1.3, 1.5) \\
x_4 &\sim \mathcal{U}(2.25, 2.55) \\
x_5 &\sim \mathcal{U}(0.4, 0.7) \\
x_6 &\sim \mathcal{U}(-0.2, 0.2) \\
x_7 &\sim \mathcal{U}(0.3, 0.55)
\end{aligned}
$$

The dynamics are given by

$$
\begin{aligned}
x_1 &= x_1 + 0.14x_3 - 0.09x_1 \\
x_2 &= x_2 + 0.25x_5 - 0.15x_2 \\
x_3 &= x_3 + 0.06x_7 - 0.08x_2x_3 \\
x_4 &= x_4 + 0.2 - 0.13x_3x_4 \\
x_5 &= x_5 + 0.07x_1 - 0.1x_4x_5 \\
x_6 &= x_6 + 0.03x_1 - 0.31x_6 \\
x_7 &= x_7 + 0.18x_6 - 0.15x_2x_7
\end{aligned}
$$

**1D Lattice Model:** This model involves 10 particles in a Lattice with a force between adjacent particles described by a double well potential function [29]. The state variables are

$$(u_1, \ldots, u_{10})$$

The initial conditions are:

$$
\begin{aligned}
u_1 &\sim \mathcal{U}(-0.5, -0.3) \\
u_2 &\sim \mathcal{U}(0.4, 0.5) \\
u_3 &\sim \mathcal{U}(-0.2, 0) \\
u_4 &\sim \mathcal{U}(-0.2, 0) \\
u_5 &\sim \mathcal{U}(0.55, 0.75) \\
u_6 &\sim \mathcal{U}(0.1, 0.3) \\
u_7 &\sim \mathcal{U}(0.55, 0.75) \\
u_8 &\sim \mathcal{U}(-0.19, 0.19) \\
u_9 &\sim \mathcal{U}(-0.6, -0.4) \\
u_{10} &\sim \mathcal{U}(-0.19, 0.19)
\end{aligned}
$$

The dynamics are given by:

$$
\begin{aligned}
u_1' &= u_1 + 0.1(0.1(u_2 - u_1) - u_1(u_1 - 1)(u_1 - 0.6)) \\
u_2' &= u_2 + 0.1(0.1(u_1 + u_3 - 2u_2) - u_2(u_2 - 1)(u_2 - 0.6)) \\
u_3' &= u_3 + 0.1(0.1(u_2 + u_4 - 2u_3) - u_3(u_3 - 1)(u_3 - 0.6)) \\
u_4' &= u_4 + 0.1(0.1(u_5 + u_3 - 2u_4) - u_4(u_4 - 1)(u_4 - 0.6)) \\
u_5' &= u_5 + 0.1(0.1(u_6 + u_4 - 2u_5) - u_5(u_5 - 1)(u_5 - 0.6)) \\
u_6' &= u_6 + 0.1(0.1(u_7 + u_5 - 2u_6) - u_6(u_6 - 1)(u_6 - 0.6)) \\
u_7' &= u_7 + 0.1(0.1(u_8 + u_6 - 2u_7) - u_7(u_7 - 1)(u_7 - 0.6)) \\
u_8' &= u_8 + 0.1(0.1(u_9 + u_7 - 2u_8) - u_8(u_8 - 1)(u_8 - 0.6)) \\
u_9' &= u_9 + 0.1(0.1(u_{10} + u_8 - 2u_9) - u_9(u_9 - 1)(u_9 - 0.6)) \\
u_{10}' &= u_{10} + 0.1(0.1(u_9 - u_{10}) - u_{10}(u_{10} - 1)(u_{10} - 0.6))
\end{aligned}
$$

## B   Missing Proofs

**Lemma 1** (Soundness of Multiplication)**.** *For given polynomial forms* $(p_1 + I_1)$ *and* $(p_2 + I_2)$, $[\![(p_1 + I_1) \otimes (p_2 + I_2)]\!] \supseteq \{f_1 \times f_2 \mid f_1 \in [\![p_1 + I_1]\!], \ f_2 \in [\![p_2 + I_2]\!] \ \}$.

*Proof.* Let $f_1 : p_1 + g_1 \in [\![p_1 + I_1]\!]$ and $f_2 : p_2 + g_2 \in [\![p_2 + I_2]\!]$. By definition of $[\![p_j + I_j]\!]$, we conclude that $g_1 \in I_1$ and $g_2 \in I_2$. Therefore $f_1 \times f_2 = p_1p_2 + p_1g_2 + p_2g_1 + g_1g_2 = p_1p_2 + g_{1,2}$ wherein $g_{1,2} = p_1g_2 + p_2g_1 + g_1g_2$. To complete the proof, we verify that $p_1p_2 + g_{1,2} \in [\![(p_1 + I_1) \otimes (p_2 + I_2)]\!]$. $\qquad\square$

**Lemma 2.** *For a* $m$ *times differentiable function* $g$ *and* $j + 1 \leq m$, *then* $[\![g(p + I)]\!] \subseteq [\![\hat{p} + \hat{I}]\!]$.

*Proof.* From the derivation of $\hat{p} + \hat{I}$, we note that $\hat{p}$ denotes the first $j$ terms of the Taylor series for $g(\mathbf{w})$ centered around $\mathbf{w} = c$. The interval $\hat{I}$ is the Lagrange remainder for this Taylor series. The proof of interval containment is directly obtained from the derivation of the Taylor series and Lagrange remainder. □

**Lemma 3.** *For any $j \in \mathbb{N}$, the moment $\mathbb{E}(\hat{f}^j)$ is bounded by the interval $I^j$.*

*Proof.* The moment can be written as an integral $\int \hat{f}^j(\mathbf{w})\mathbb{P}(d\mathbf{w})$. $\hat{f}^j$ is a continuous function and $\mathbb{P}(d\mathbf{w})$ is a measure over a compact set. Thus, the integral can be shown to exist. Let $J : [l_j, u_j]$ denote the interval $I^j$. We note that $l_j \int \mathbb{P}(d\mathbf{w}) \leq \int \hat{f}^j(\mathbf{w})\mathbb{P}(d\mathbf{w})$. Also, $\int \hat{f}^j(\mathbf{w})\mathbb{P}(d\mathbf{w}) \leq u_j \int \mathbb{P}(d\mathbf{w})$. Combining, we have $\int \hat{f}^j(\mathbf{w})\mathbb{P}(d\mathbf{w}) \in [l_j, u_j]$. □

## C   Balanced Partitioning of Polynomial Forms

Let the polynomial $p := \sum_{j=1}^m a_j \mathbf{w}^{\mathbf{r}_j}$, where $\mathbf{w}^{\mathbf{r}_j}$ is a power product with degree $\mathbf{r}_j$. We have previously shown how the polynomial can be split into mutually independent components $p = p_1 + \ldots + p_J$, wherein each $p_i$ involves a subset $W_i \subseteq W$ of the noise symbols and $W_i \cap W_j = \emptyset$ for $i \neq j$. This is achieved through a strongly connected component (SCC) decomposition of a graph whose nodes are the noise symbols in $W$ and whose edges include $(w_i, w_j)$ if there exists a term in $p$ that involves $w_i$ and $w_j$ with strictly positive powers.

We can extend this approach even further to combine splitting with polynomial truncation. The key idea is to select some of the terms $\mathbf{w}^{\mathbf{r}_j}$ from $p$ and truncate these terms by removing them from $p$ while adding them to the error interval. We wish to satisfy two objectives in doing so:

1. Split the polynomial $p$ into a larger number of mutually independent components $L > J$;
2. At the same time, we wish to minimize the growth in the width of the error interval due to truncation.

We briefly describe a reduction to the balanced partitioning problem over graphs. For convenience, we will assume that the noise symbols of $p$ all belong to a single strongly connected component. I.e, the algorithm is achieved to split each "large" component $p_l$ of $p$ for $l \in [1, J]$.

$(K, L)$ **balanced partitioning:**   Given a weighted graph $G$ of size $n$ and a numbers $0 < K < n$ and $0 < L < \frac{n}{K}$, we wish to partition the vertices of the graph into $K$ mutually disjoint subsets wherein (a) each component has at least $L$ vertices in it, and (b) the sum of edge weights for the edges whose vertices lie in different components is minimized. It is well-known that the $(K, L)$ balanced partitioning problem is NP-hard. On the other hand, Andreev and Räcke report a $O(\log^2(n))$ factor approximation for a very closely related formulation under the assumption that $L < \alpha \frac{n}{K}$ for $\alpha < 1$ [3].

The process of splitting a polynomial $p$ can be seen as a balanced partitioning problem. Once again, recall that $p := \sum_{j=1}^m a_j \mathbf{w}^{\mathbf{r}_j}$, where $\mathbf{w}^{\mathbf{r}_j}$ is a power product with degree $\mathbf{r}_j$. First, given a polynomial $p$ with noise symbols $W$, we create a weighted graph with vertices in $W$. For each $(w_i, w_j)$ $(i \neq j)$ there is an undirected edge if $w_i, w_j$ occur together in a term $\mathbf{w}^{\mathbf{r}_k}$ of the polynomial with strictly positive powers. The weight of the edge $w_{i,j}$ is taken as follows:

$$\sum \left( \{|\mathsf{range}(a_k \mathbf{w}^{\mathbf{r}_k})| \text{ s.t. } \mathbf{r}_{k,i} \geq 1 \text{ and } \mathbf{r}_{k,j} \geq 1\} \right) .$$

Here range of a term refers to its range computed as an interval knowing the ranges of each noise symbol $w_i$ and for an interval $I : [\ell, u]$ its width $|I|$ is given by $u - \ell$.

Once the graph is setup, we use a standard mixed integer formulation of the graph partitioning problem. This formulation involves binary variables $x_{i,k}$ that indicates whether vertex $i$ is placed in partition $k$ for each vertex $i$ and for each $k \in [1, K]$. Likewise, we use an indicator variable $w_{i,j}$ for each edge to denote whether it spans two different partitions. While this can be prohibitively expensive in practice, we can adjust the complexity by placing a limit on how much error we are willing to tolerate in order to eliminate terms from consideration. This, in turn, reduces the size of the graph by eliminating vertices and edges from consideration.

Table 2: Results on nonlinear benchmark model comparing our approach with and without the balanced partitioning (BP) scheme. $N$: # time steps, $D$: degree of polynomial form. All timings indicated by the ⏱ symbol are in seconds and split between time taken to compute polynomial forms versus time to compute queries (includes time to compute balanced partitions).

| Benchmark | $N$ | $D$ | w/o BP | w/ BP |
|---|---|---|---|---|
| Cartpole | 8 | 2 | ⏱[45.1s, 1.4s] | ⏱[44.6s, 1.2s] |
| | | $\mathbb{E}(x)$: | [-0.189,0.2075] | [-0.189,0.2075] |
| | | $\mathbb{E}(\theta)$ : | [-0.354,0.357] | [-0.354,0.357] |
| | $\mathbb{P}(x \geq 2)$ : | | | |
| | | C-H | 0.1664 | 0.1276 |
| | | Cb | 0.0065 | 0.0065 |
| | | B | 0.066 | 0.055 |
| | $\mathbb{P}(\theta \geq \pi/6)$ : | | | |
| | | C-H | 0.4711 | 0.4573 |
| | | Cb | 0.0198 | 0.0198 |
| | | B | 0.1635 | 0.1575 |

Table 3: Cart-pole model: polynomial form performance with fixed degree D=2 and time steps N=8 for various number of cluster of balanced partitioning K. $T_q$ is the time taken to compute the result of queries, which includes solving mixed-integer optimization problem for balanced partitioning. $N_p$ is the number of split polynomials and $N_l$ is the largest size split component (number of terms).

| K | $T_q$ | $N_p$ | | $N_l$ | | $\mathbb{P}(x \geq 2)$ | | | $\mathbb{P}(\theta \geq \pi/6)$ | | |
|---|---|---|---|---|---|---|---|---|---|---|---|
| | | $x$ | $\theta$ | $x$ | $\theta$ | **C-H** | **Cb** | **B** | **C-H** | **Cb** | **B** |
| 3 | 1.2 | 11 | 14 | 92 | 68 | 0.1276 | 0.0065 | 0.055 | 0.4573 | 0.0198 | 0.1575 |
| 4 | 2.2 | 12 | 15 | 89 | 66 | 0.1092 | 0.0065 | 0.049 | 0.4457 | 0.0198 | 0.1525 |
| 5 | 3.6 | 13 | 16 | 86 | 63 | 0.1014 | 0.0065 | 0.047 | 0.4290 | 0.0198 | 0.1454 |
| 6 | 6.4 | 14 | 17 | 83 | 60 | 0.0976 | 0.0065 | 0.046 | 0.4162 | 0.0198 | 0.1401 |
| 7 | 38.9 | 15 | 18 | 80 | 58 | 0.0918 | 0.0065 | 0.044 | 0.4070 | 0.0198 | 0.136 |
| 8 | 2277 | 16 | 19 | 78 | 55 | 0.0756 | 0.0065 | 0.039 | 0.4037 | 0.0198 | 0.135 |

**Performance** We now report on a preliminary evaluation for this scheme over a subset of our benchmark set using the GLPK solver to compute balanced partitions. The cart-pole model will be used in our evaluation to illustrate the idea of splitting by dependency reduction. Table 2 shows that using balanced partitioning with $K = 3$ provides the same Chebyshev bounds but smaller Chernoff-Hoeffding bounds and Bernstein bounds than those without using balanced partitioning. Also, Hoeffding bounds and Bernstein bounds can be reduced as increasing the number of cluster $K$, which is shown in Table 3. However, because there is a trade-off between the accuracy and the computation time of solving a mixed-integer LP problem, it becomes very time consuming as $K = 8$ in the cart-pole model. Also, we notice that the number of split polynomials $N_p$ does not increase significantly with increasing $K$.