[Reviews · NeurIPS 2020]

Review 1

Summary and Contributions: This paper presents a presentation of the state variable's distributions as polynomial forms. The authors further elaborate on the properties of this representation and show that the polynomial form is closed under elementary operations. This subsequently allows one to bound the expectations and moments under certain regularity assumptions.

Strengths: I like the general idea of this work and how the proposed approach generalizes previous work that was restricted to affine functions. The major benefit is, that this allows one to account for nonlinearities in dynamical systems. Therefore, one can expect that bounds on tail probabilities will be much tighter when using the representation as polynomial form. For a couple of benchmark examples, it is shown that this is often the case.

Weaknesses: While the representation of the distributions as a polynomial form is very general, it introduces a notable computational overhead. This problem is already mentioned in Example 3 and becomes evident when looking at the experimental results (that consequently only consider a few time steps) in Table 1. I am afraid that this limits the practical applicability of the proposed method considerably. One should keep in mind that we are estimating the uncertainties in dynamical systems. In this context (e.g. applications in control systems or tracking) one usually wants to reason about the uncertainties during run-time and not ex-post. This is the very reason why scalable algorithms that rely on message-passing (e.g., all variants of Kalman filters) are prevalent in this field. At least I would expect that this limitation is mentioned and discussed.

Correctness: The theoretical results seem to be correct. The empirical evaluation is done on a couple of well-established benchmark nonlinear dynamical systems.

Clarity: The paper is well written. I particularly enjoyed the logical structure and the expressive examples. This makes the paper relatively accessible even for readers not too familiar with the background literature.

Relation to Prior Work: Honestly, I am not too familiar with the corresponding literature but -- from what I can tell -- prior work is mentioned adequately and is clearly stated in what respect the current paper extends prior work.

Reproducibility: Yes

Additional Feedback: I have a couple of questions and minor comments: - As mentioned above, I do have concerns about the scalability. In Example 3 we can see how a very simple form x(1) is already grown substantially for x(20). Is it possible to make some statements about how fast x(t) grows, given a specific model? - When evaluating the expectation during the computation of the moments it is necessary to decompose the large polynomial into independent components. I suppose that this is necessary because of the many cross-terms that occur in higher-order moments. Wouldn't it make sense to consider orthogonal descriptors, i.e., cumulants instead? I guess this could already lower the computational burden a bit. Comments: - There is no distinction in notation between state variables (e.g., x) and random variables (e.g., w_1) and both are simply referred to as variables. I would prefer if this distinction is made more prominent. - The equation in Example 1 stands out of the text-body; this could easily be fixed using some negative \hspace*. Also, this is the only time that the random variables are depicted in red. I would prefer them to be black for the sake of consistency. - line 106-109: I do not fully agree. If all distributions belong to an exponential family it is often tractable to reason about the distributions (cf. [R1,R2]). In particular, if everything is Gaussian (which -- I agree -- need not always be the case) all conditionals and marginals remain Gaussians. --- Edited after reading the author's response --- I have read the author's response, in which they address the two most important issues I have with the paper in its current form. First, on the growth of the number of terms: the authors explain how fast the number of terms grows and argue how they can limit the growth by adjusting the threshold. Although this is still somewhat vague, experiments are promised that show how the threshold affects the number of terms. Second, regarding the question how to make polynomial forms work during runtime. To do so, the authors argue that their approach can be extended to some form of message passing. I agree that this would be a viable option if one maps the polynomial form back to a distribution again. I am, however, not convinced that this is as straightforward as claimed as approximating distributions by a truncated series is almost always problematic as long as we don't restrict the type of distributions. Overall, I believe that the proposed approach still has its limitations but also has its merits. Given that the paper is well written and easy to follow it may inspire future work.


Review 2

Summary and Contributions: This paper proposes a method to use polynomial forms to represent the distribution of state variables in discrete-time stochastic dynamical systems and provide guaranteed bounds on the distributions of the state variables. The main idea is to combine higher order interval arithmetic and concentration of measure inequalities to quantify uncertainty. This work is built upon the author’s previous work that uses affine forms rather than polynomial forms to quantify uncertainty. ------------------------------------------------------------------------------------------------------------ After the reading the rebuttal, I agree with the author that P-box scales poorly in the intended applications of this method, thus I'm fine without comparing to it. The comparison with Boussiou et al's method in Table 1 shows the proposed method generates a much tighter bound, while the polynomial method generally requires a lot more computation time. Therefore, I decide to increase my score to marginally above acceptance.

Strengths: The paper provides very-detailed description and walk-through examples of the proposed method that will help to better understand the technical methods. The proposed method provides more rigorous bounds on the uncertainty of stochastic dynamical systems due to the influence of the uncertainties in the initial state and the stochastic disturbance inputs.

Weaknesses: The significance of the paper is limited due to the following reasons: 1) A more thoroughly evaluation is needed. This paper only provides comparison in empirical evaluation with the method based on affine-forms. How about performance against other methods, such as P-box [1] 2) A more in-depth comparison with the closely related work [2] is needed. see "relation to prior work" [1] Ferson, Scott, et al. Constructing probability boxes and Dempster-Shafer structures. No. SAND-2015-4166J. Sandia National Lab.(SNL-NM), Albuquerque, NM (United States), 2015. [2] Olivier Bouissou, Eric Goubault, Sylvie Putot, Aleksandar Chakarov, and Sriram Sankaranarayanan. Uncertainty propagation using probabilistic affine forms and concentration of measure inequalities. In Tools and Algorithms for Construction and Analysis of Systems (TACAS), volume 9636 of Lecture Notes in Computer Science, pages 225–243. Springer-Verlag, 2016.

Correctness: The claims and method seems correct.

Clarity: The paper is generally readable with some minor grammar mistakes.

Relation to Prior Work: The work of uncertainty quantification is mass and the author discussed several closely related work. Since the work extends the author's previous work (as stated by the author) that uses affine forms, it’ll be very helpful if the authors could explicitly discuss either the advantage of using polynomial forms rather than affine forms or the difference in the application scenarios of the two methods.

Reproducibility: Yes

Additional Feedback:


Review 3

Summary and Contributions: The authors study models of discrete-time stochastic dynamical systems. They use polynomial models combined with interval arithmetic. The variables of the polynomials are random variables and the intervals allow them to formally bound uncertainties. This allows them to attain impressively good approximations by a formally-correct method.

Strengths: The claims appear to be sound. The use of interval arithmetic combined with Taylor series provides a very general method.

Weaknesses: Not surprisingly, in experiments their high-degree polynomial model provides tighter bounds than those obtained from an affine (degree-1 polynomial) model. Naturally, this incurs greater computational cost. No theoretical bounds on time or space complexity of the method are given.

Correctness: Yes

Clarity: Yes

Relation to Prior Work: Yes

Reproducibility: Yes

Additional Feedback: Minor comments and questions: line 20: say what a UAV is. line 28-29: Avoid inverted commas in an academic paper. I have no idea what you mean by “nondeterministic” uncertainties. What is the difference between nondeterministic and deterministic uncertainty? line 32: forms lines 52-55: The sentence 'In the probabilistic case...' has no verb (or an extra 'that'). line 78: why n times 1? Do you mean n times m (where m is the number of parameters required to describe the state). line 103: P is probably in the wrong typeface. Shouldn't it be the same as in the following equation to disnguish it from the distribution function P? line 109: delete 'whereas' line 180: By 'minimum' do you mean 'maximum'? You choose to keep the details of the low degree terms and treat the high-degree terms as uncertainty. This only made sense to me a few lines later when you refer to Taylor series, so you should perhaps prepare the reader for this. line 275: compute utilize: extra word? line 289: missing verb (or extra 'that') in the sentence 'The constants..' Table 1: Ebola: i or I? Bibliography: to enforce capital letters in chebyshev, congo, uganda, taylor, mpfi, ode, use {C}hebyshev, {C}ongo, etc. Questions: (1) Are there not other techniques you could have compared with other than the affine method? (2) An obvious question is what is the theoretical computational complexity (for example with a bound of d on degree, for n time steps, m parameters) of the proposed method? Is the number of terms potentially m^d? In example 3, there are 42 noise symbols (parameters) and 269 terms after 20 time steps, but this is just one example. The reader would like to be given a theoretical upper bound on the number of terms.

[Author Response · NeurIPS 2020]

We thank the reviewers for the detailed comments and valuable feedback. We have paraphrased questions from the reviewers and provided answers. We are particularly thankful to the reviewers for pointing out typos, and providing numerous suggestions for improvement. We will fix these in our revised version.

**Reviewer # 1:** How fast does the size of the polynomial form $x(t)$ grow? **Reviewer # 3:** Apropos theoretical upper bound on the number of terms.

As reviewer # 3 points out, if we restrict ourselves to a degree at most $d$ polynomial over $m$ variables, the number of terms grows as $O((n|W|)^d)$ where $n$ is the number of time steps, $|W|$ is the number of disturbance variables in the system and $d$ is the degree limit. In our implementation, we substantially reduce this number by truncating terms whose bounds are below a threshold: we add these terms to the residual interval. This makes our polynomials smaller. We could increase this threshold tolerance to adjust the polynomial size further: we hope to report on these effects as part of our extended version of our evaluation.

**Reviewer # 1:** Wouldn't it make sense to consider orthogonal descriptors, i.e., cumulants instead?

This is a very nice suggestion, thank you! Though we do not use cumulants, we use a trick described in lines 268-272 to reduce the overhead of computing the first four central moments of the sum of the individual components $p_1, \ldots, p_j$. The first three central moments are directly computed by adding the individual central moments (because they coincide with the cumulants!!) but the fourth central moment formula is slightly more involved since it involves the pairwise products of variances. However, with cumulants, we can simply add them. This would lead to very nice computational savings, much like the FFT polynomial multiplication algorithm. The key, however, is to compute the cumulants of each $p_j$ rapidly.

**Reviewer #1**: one usually wants to reason about the uncertainties during run-time and not ex-post. This is the very reason why scalable algorithms that rely on message-passing (e.g., all variants of Kalman filters) are prevalent.

This is a good observation and we will clarify in our revision as follows: Polynomial forms accumulate the effect of disturbances from the very first step to the current step. (Extended) Kalman filters can be seen as using degree 1 terms (affine terms) and "compressing the distribution" from time 1 to n-1 into a single multivariate Gaussian before tackling time n. In some sense, our approach naturally lends itself to this kind of message passing if one is willing to approximate the polynomial form by a distribution at time $n - 1$ before tackling the state update. Note that in filtering, there is the uncertainty propagation and the conditioning part. Conditioning on observations is not in scope for this paper but nevertheless an interesting problem for future work.

**Reviewer # 2**: Comparison with work of Scott Ferson et al. and the p-box method [12]

A p-box is a discretization of the joint distribution over the state space into cells with upper and lower bounds for the measure in each cell. Ferson's approach uses p-boxes as its representation of probability distribution. It allows us to answer queries about assertions and moments by summing up the contribution from each tile. As one may imagine, beyond 2 or 3 dimensions the number of tiles needed to maintain an accurate enough p-box representation to answer probability queries effectively can become prohibitive. Note that a discretization of the disturbances at each step is needed as well and the approach will effectively do an expensive convolution over all the tiles for the state variables and disturbances. Also note that five of the larger benchmarks in Table 1 have 5-10 state variables.

**Reviewer # 2**: Comparison with Bouissou et al. **Reviewer # 2**: Question about comparison with related work.

Please see Table 1 for the comparison with Bouissou et al. As you can see, polynomial forms provide much tighter bounds for moment and probability queries. Although one pays the price for using polynomials and higher order terms, polynomial forms are even faster in a few instances: this is because affine forms of Boussiou et al model nonlinear terms by fresh noise symbols, tracks correlation between the noise symbols and even maintains an ever growing covariance matrix involving the noise symbols. All of these are avoided in our polynomial forms approach.

**Reviewer # 3** Are there not other techniques you could have compared with other than the affine method?

The only other method we could have compared apples to apples is perhaps the approach of Ferson et al with suitable alternations to fit our problem setup. However, we do not expect p-boxes to be scalable or precise (see response to Reviewer # 2 on comparison with Ferson et al. [12]). There are methods that are based on Monte-Carlo simulations, linearization of the dynamics, truncation and such: although we have cited some of these, we did not compare against them since those techniques have different formal guarantees. We have provided results from $10^5$ simulations for reference.

[Meta-Review · NeurIPS 2020]

This paper proposes a method to use polynomial forms to represent the distribution of state variables in discrete-time stochastic dynamical systems and provide guaranteed bounds on the distributions of the state variables. The main idea is to combine higher order interval arithmetic and concentration of measure inequalities to quantify uncertainty. This work builds upon the authors previous work that uses affine forms rather than polynomial forms to quantify uncertainty. All reviewers found the paper clearly written and are positively predisposed towards acceptance. One reviewer raised two points relating to comparison to related methods and raised their score after the authors addressed these points in their response. Another reviewer noted post response that while the paper has limitations but it also has merits. As long as the camera ready adequately discusses these limitations, this paper may inspire future work.